# Alzheimer’s Disease Related Biomarkers Were Associated with Amnestic Cognitive Impairment in Parkinson’s Disease: A Cross-Sectional Cohort Study

**DOI:** 10.3390/brainsci14080787

**Published:** 2024-08-02

**Authors:** Xiaofan Xue, Shanshan Mei, Anqi Huang, Zhiyue Wu, Jingrong Zeng, Haixia Song, Jing An, Lijuan Zhang, Guozhen Liu, Lichun Zhou, Yanning Cai, Baolei Xu, Erhe Xu, Piu Chan

**Affiliations:** 1Department of Neurology and Neurobiology, Xuanwu Hospital of Capital Medical University, Beijing 100053, China; pauline_xuexiaofan@126.com (X.X.); meishanshan@xwhosp.org (S.M.); anchop98@163.com (A.H.); 18752536088@163.com (Z.W.); zengjingrong96@sina.com (J.Z.); baolei_x@163.com (B.X.); xuerhe@xwhosp.org (E.X.); 2Department of Neurology, Beijing Chaoyang Hospital, Capital Medical University, Beijing 100043, China; zhoulichun@bjcyh.com; 3Department of Neurology, The People’s Hospital of Shijiazhuang, Shijiazhuang 050000, China; xiaoxixia2022@163.com; 4Department of Neurobiology, Xuanwu Hospital of Capital Medical University, Beijing 100053, China; anjing@xwhosp.org; 5National Clinical Research Center for Geriatric Diseases, Xuanwu Hospital of Capital Medical University, Beijing 100053, China; lijuancn_wang@163.com; 6Parkinson’s Disease Cloud Medical Technology Company, Beijing 100055, China; liuguozhen@pdhospital.com; 7Department of Clinical Biobank and Central Laboratory, Xuanwu Hospital of Capital Medical University, Beijing 100053, China; caiyanning@xwhosp.org

**Keywords:** Parkinson’s disease, cognitive impairment, biomarkers

## Abstract

Background: Cognitive impairment is common in patients with Parkinson’s disease (PD) and occurs through multiple mechanisms, including Alzheimer’s disease (AD) pathology and the involvement of α-synucleinopathies. We aimed to investigate the pathological biomarkers of both PD and AD in plasma and neuronal extracellular vesicles (EVs) and their association with different types of cognitive impairment in PD patients. Methods: A total of 122 patients with PD and 30 healthy controls were included in this cross-sectional cohort study between March 2021 and July 2023. Non-dementia PD patients were divided into amnestic and non-amnestic groups according to the memory domain of a neuropsychological assessment. Plasma and neuronal EV biomarkers, including α-synuclein (α-syn), beta-amyloid (Aβ), total tau (T-tau), phosphorylated tau181 (p-tau181), and glial fibrillary acidic protein (GFAP), were measured using a single-molecule array and a chemiluminescence immunoassay, respectively. Results: Neuronal EV but not plasma α-syn levels, were significantly increased in PD as compared to healthy controls, and they were positively associated with UPDRS part III scores and the severity of cognitive impairment. A lower plasma Aβ42 level and higher neuronal EV T-tau level were found in the amnestic PD group compared to the non-amnestic PD group. Conclusions: The results of the current study demonstrate that neuronal EV α-syn levels can be a sensitive biomarker for assisting in the diagnosis and disease severity prediction of PD. Both AD and PD pathologies are important factors in cognitive impairment associated with PD, and AD pathologies are more involved in amnestic memory deficit in PD.

## 1. Introduction

Cognitive impairment is a common non-motor symptom of Parkinson’s disease (PD), and is associated with poor quality of life. PD patients have a higher risk of developing dementia (PDD), with the incidence of PDD affecting 46% of patients with a more than 10-year history of PD [1]. Previous studies have found that α-syn toxicity in PD can result in neuronal apoptosis and cognitive dysfunction [2]. In addition, the pathology of Alzheimer’s disease (AD), the most common neurodegenerative disease of cognitive dysfunction in older people, has been found to be comorbid in 28.6% of PD patients [3]. Extensive studies have shown a relationship between AD-related biomarkers (β-amyloid (Aβ) 42, Aβ42/Aβ40 ratio, total tau (T-tau), phosphorylated tau181 (p-tau181), and glial fibrillary acidic protein (GFAP)) and cognitive impairment in patients with PD [4,5,6,7]. However, it is still controversial as to whether PD and AD pathologies are comorbidly or independently associated with cognitive impairment in PD. Their application as biomarkers for the prediction and diagnosis of cognitive impairment remains unclear.

Changes in α-syn, the Aβ42/Aβ40 ratio, and tau levels have been shown to predict cognitive decline and the time to dementia in PD, showing potential as prognostic biomarkers [4,5,8,9,10]. In general, cerebrospinal fluid (CSF) would be the best, but its accessibility and application are limited. Plasma and extracellular vesicle (EV) levels of pathological biomarkers are commonly used, but inconsistent results have been obtained [2,11,12,13]. Neuronal EVs have been demonstrated to more closely represent the micro-environmental condition in the brain [14] because they can cross the brain–blood barrier without the loss of their integrity due to the lipid membrane structure [14]. Neuronal EVs can carry the molecular information of neurons in terms of the inner protein and nucleotide profile to the periphery [14].

As a predictor for the conversion to PDD, PD mild cognitive impairment (PD-MCI) is of utmost clinical relevance and spans multiple cognitive domains [15,16,17]. Whether the risks of the diverse subtypes of PD-MCI and the progress toward PDD are different is still unclear. In some prospective studies, lower baseline concentrations of Aβ42 were associated with memory impairment but not with executive–attentional or visuospatial dysfunction [15,16]. However, the PD patients analyzed in the above studies were not divided into different subtypes, which may potentially affect the results. One previous study indicated that the amnestic PD-MCI subtype showed the lowest performance in global cognitive scores, indicating that this subtype might be the closest to developing PDD [18]. Interestingly, elevated cortical Aβ and tau were found in more than 67% of PDD patients in postmortem studies [19], demonstrating that AD-related proteins might play an essential role in PDD. However, limitations and controversy still surround the issue of whether these AD-related biomarkers can discriminate individuals with amnestic PD cognitive impairment from a non-amnestic group and predict the risk of conversion to PDD.

In light of the above, this study aimed to compare six biomarkers (α-syn, Aβ42, Aβ42/Aβ40 ratio, T-tau, p-tau181, and GFAP) using plasma and neuronal EVs between HC and different levels of cognitive function in patients with PD in order to find a biomarker that can provide meaningful information about movement or cognitive severity. Meanwhile, we compared different biomarkers between amnestic and non-amnestic PD non-dementia patients, to provide insights into the relationship between different biomarkers and the amnestic subtypes of PD patients. This is the first study to evaluate the value of different biomarkers for the diagnosis of amnestic subtype in PD patients.

## 2. Materials and Methods

### 2.1. Participants and Clinical Assessments

In this cross-sectional cohort study, we recruited 122 consecutive patients with PD admitted to the Clinical and Research Center of Xuanwu Hospital of Capital Medical University in Beijing, China, between March 2021 and July 2023. All patients were diagnosed as clinical or probable PD according to the Movement Disorder Society (MDS) Diagnostic Criteria for Parkinson’s Disease [20] by movement disorder specialists. The exclusion criteria were (1) a history of serious diseases affecting brain function, including cerebral vascular diseases (Fazakes ≥ II; Hachinski > 4), cerebral trauma, encephalitis, epilepsy, etc.; (2) a history of other serious diseases affecting cognitive function, including severe mental health diseases, cancer, hyperthyroidism, carbon monoxide poisoning, intemperance, drug addiction, and severe systemic diseases; (3) poor cooperation, including auditory and visual impairment, aphasia, and severe paralysis; and (4) a history of taking anticholinergic drugs. The motor symptoms were evaluated using the MDS Unified PD Rating Scale (MDS-UPDRS) Part 2/3 scores and disease severity, using the Hoehn–Yahr (H-Y) stages. The MoCA was used to assess cognitive impairment and diagnose MCI using a cutoff score of <26 with a 90% sensitivity and 75% specificity [21]. In addition, the Hamilton Depression Scale (HAMD) was used to assess the severity of depression, and the Epworth Sleepiness Scale (ESS) was used to assess the severity of sleepiness. We also enrolled 30 sex- and age-matched healthy controls (HCs) with a MoCA score greater than 25.

### 2.2. Neuropsychological Evaluations

All participants were subjected to a standard battery of clinical and neuropsychological assessments. According to the MDS recommendation, the following memory cognitive domain tests were evaluated: the Rey Auditory Verbal Learning Test (AVLT) and the Wechsler Logical Memory Test (LMT) (taking the average score of three assessments of immediate recall memory, as well as the score for short-term and delayed recall memory). Neuropsychological performance was considered impaired when the subject scored a standard deviation (SD) of 1.96 below the normality cut-off values [16,22]. PD patients without dementia were defined as follows: amnestic normal cognitive PD (PD-NC), with a total MoCA score ≥ 26 and impairment on one of the memory tests; non-amnestic PD-NC, with a total MoCA score ≥ 26 and without the involvement of the memory domain; amnestic PD-MCI, with a total MoCA score ≥ 21 but less than 26 and impairment on at least one of the memory tests; non-amnestic PD-MCI, with a total MoCA score ≥ 21 but less than 26 and without the involvement of the memory domain.

### 2.3. Sample Collection and Assay of Plasma Biomarkers

We collected a 10 mL venous blood sample from each patient after more than 6 h of fasting. The samples were centrifuged at 1500× *g* for 10 min and immediately stored at −80 °C until testing. The levels of Aβ42, Aβ40, T-tau, p-tau181, and GFAP were determined via a single-molecule array (Simoa, Quanterix, Boston, MA, USA), using a Simoa Human Neurology 4-Plex E kit and a Simoa p-tau181 V2 Advantage kit (Quanterix, Billerica, MA, USA) according to the manufacturers’ protocols. All the plasma samples were tested in duplicate. For the detection of plasma α-syn, the samples were diluted 100-fold with Phosphate-Buffered Saline (PBS) to adjust the protein concentration to an appropriate range for detection. Subsequently, the diluted aliquots were analyzed using a commercial assay kit (B-001, Kaixianghongkang, Beijing, China) via the chemiluminescence immunoassay (CLIA) method.

### 2.4. Enrichment of Neuronal EVs and Assay of Biomarkers

A commercially available kit (specifically the Kaixianghongkang neuronal EV extraction kit, C-005, Kaixianghongkang, Beijing, China) was used to isolate neuronal EVs from plasma samples, strictly following the procedural instructions provided by the manufacturer. Under the guidance of MISEV2023, neuronal EVs were characterized using a nanoparticle tracking analysis (NTA), Western blotting (WB), and transmission electron microscopy (TEM), respectively. The instrument utilized for this procedure was a NanoSight NS300 (Malvern Panalytical, London, UK). Data acquisition and analysis were carried out using NTA software version 3.4.

To verify the presence of typical EV markers in neuronal EVs, a Western blotting (WB) analysis was performed. A 10% sodium dodecyl sulfate–polyacrylamide gel electrophoresis (SDS-PAGE) gel was prepared using a commercial kit (Biotides, Beijing, China). The neuronal EV lysate was boiled for 5 min before loading 10 μL onto the gel for electrophoresis. Following electrophoresis, the proteins were transferred to a polyvinylidene difluoride (PVDF) membrane, which was then blocked with a quick protein blocking solution for 5–10 min. After blocking, the membrane was incubated overnight at 4 °C with specific primary antibodies, including anti-TSG101 and anti-TUJ1. The membrane was washed and then incubated with the corresponding horseradish peroxidase-conjugated secondary antibodies. Detection was performed using an ECLTM-enhanced chemiluminescence system (Epizyme, Shanghai, China) and imaged with a chemiluminescence gel documentation system (Touch Imager, eBlot, Nanjing, China) (Appendix A).

A JEM-1400 platform (JEOL, Akishima, Japan) was utilized to examine the structural integrity and morphological characteristics of the neuronal EVs. The results for the neuronal EV biomarkers were finally adjusted through the cluster of differentiation (CD) 81 using the CLIA method.

### 2.5. Statistical Analysis

Statistical analyses were constructed using IBM SPSS 25.0. Diagrams were performed GraphPad Prism 9.5.0. Continuous variables were expressed as mean ± SD or median (interquartile range). Continuous variables were compared using *t*-test or Mann–Whitney U test. Categorical variables were presented as percentages and analyzed using chi-square tests and a Fisher’s exact test. Correlations between clinical characteristics and the levels of periphery blood biomarkers were assessed using Spearman’s correlation analysis. The discriminatory properties of plasma and neuronal EVs biomarkers for prediction of memory domain cognitive impairment were investigated using receiver operating characteristic (ROC) curves analyses of the non-amnestic PD-NC/MCI group versus the amnestic group in order to obtain cut-off values at the greatest area under the ROC curves (AUC). Two-sided *p* < 0.05 were considered significant.

## 3. Results

### 3.1. Participant Characteristics

The participants consisted of 122 PD patients including 56 PD-MCI patients, 35 PDD patients, 31 PD-NC patients, and 30 HCs. As shown in Table 1, there were no significant differences between the PD and HC groups in terms of age, sex, and education; however, these were significantly different among the PD groups with different levels of cognitive function. With an increase in the severity of cognitive decline, the patients were much older, with a lower education level and a more severe PD stage and symptoms. In addition, the percentage of males was significantly higher in the PD-MCI group.

### 3.2. The Levels of Biomarkers in HCs and PD Patients with Different Cognitive Impairment

As shown in Figure 1, there were no significant differences between the PD and HC groups in the plasma levels of α-syn, Aβ42, Aβ42/Aβ40 ratio, T-tau, and p-tau181 levels; however, the PD patients had a significantly higher level of GFAP (99.583 ± 42.135 pg/mL vs. 70.415 ± 19.539 pg/mL, *p* < 0.001) than the HCs, and the PDD patients had lower T-tau levels (0.863 ± 0.451 pg/mL vs. 1.294 ± 0.595 pg/mL, *p* < 0.01) than the PD-NC patients. As shown in Figure 2, the results of the neuronal EV levels of Aβ42, Aβ42/Aβ40 ratio, T-tau, p-tau181, and GFAP were not significantly different between the PD and HC groups except that α-syn was significantly higher in the PD group compared to the HC group (133.806 ± 93.889 vs. 69.328 ± 45.475, *p* < 0.001). Furthermore, the PD-MCI patients had a significantly higher Aβ42/Aβ40 ratio (0.342 ± 0.146 vs. 0.285 ± 0.057, *p* < 0.05) than the PD-NC patients (Appendix A). The ROC curve analyses of the plasma GFAP and neuronal EV α-syn levels for the prediction of PD versus HC are presented in Appendix A.

### 3.3. The Levels of Biomarkers in Amnestic and Non-Amnestic PD Patients

When the PD-NC patients were divided into amnestic and non-amnestic groups, no significant differences were found in the clinical characteristics (Appendix A) and the levels of plasma and neuronal EV α-syn, Aβ42/Aβ40 ratio, p-tau181, and GFAP (Table 2). We also analyzed the p-tau181/T-tau ratio as a new biomarker; however, there was no significant difference between the groups. Interestingly, the plasma Aβ42 levels were significantly lower in the amnestic than the non-amnestic group (5.579 ± 0.761 vs. 6.690 ± 1.452, *p* = 0.031 < 0.05). When the PD-MCI patients were divided into amnestic and non-amnestic groups, no significant differences were found in the clinical characteristics (Appendix A) and the levels of plasma and neuronal EV α-syn, Aβ42/Aβ40 ratio, p-tau181, and GFAP, except neuronal EV T-tau level was significantly higher in the amnestic than the non-amnestic group (0.337 ± 0.037 vs. 0.203 ± 0.095, *p* = 0.046 < 0.05) (Table 2). ROC curve analyses of the plasma Aβ42 and neuronal EV T-tau levels for predicting amnestic PD-ND versus non-amnestic PD-ND are presented in Appendix A.

### 3.4. Correlation between Clinical Characteristic and Levels of Biomarkers

Further analyses were carried out to investigate the correlation between the clinical characteristics and the levels of biomarkers. It was found that age was negatively correlated with the plasma levels’ Aβ42/Aβ40 ratio but positively correlated with the levels of GFAP. Moreover, the level of p-tau181 was positively correlated with the score of the ESS (Appendix A). In addition, the neuronal EV α-syn level was positively correlated with the UPDRS III score, and the neuronal EV T-tau level was also positively correlated with the percentage of males (Appendix A).

## 4. Discussion

As a progressive neurodegenerative disease, PD can eventually lead to physical disability and cognitive dysfunction [23]. Previous data indicated that different cognitive patterns in patients with PD may also reflect different neuropathological subtypes of PD [24,25,26]. Importantly, these patterns may be differentially related to the risk of developing PDD [27,28]. A biomarker that can accurately diagnose cognitive impairment in PD is valuable if it can provide meaningful information about the cognitive subtype or severity. However, due to the complex multifactorial nature of PD, finding a biomarker with these properties has been difficult [2]. To the best of our knowledge, this is the first study in patients with PD to evaluate the diagnostic value of different levels of plasma and neuronal EV biomarkers to identify different cognitive subtypes, especially the amnestic and non-amnestic subtypes. The major findings of this study are as follows. The results indicated that plasma GFAP and neuronal EV α-syn levels were significantly increased in the PD group compared to the HC group. A decreased plasma Aβ42 level was found in the amnestic PD-NC group compared to the non-amnestic PD-NC group, and an increased level of neuronal EV T-tau was found in the amnestic PD-MCI group compared to the non-amnestic PD-MCI group.

Pathologically, PD is characterized by the degeneration of the dopaminergic nigrostriatal system, including the overdeposition of α-syn in Lewy bodies (LBs) and Lewy neurites [29]. A number of studies have found that α-syn toxicity in PD can result in neuronal apoptosis and cognitive dysfunction [2]. A recent imaging study found that PD patients with cognitive impairment had a higher plasma level of α-syn compared with normal adults [30]. Compared with plasma α-syn, neuronal EVs contain more insoluble a-syn, oligomeric a-syn, and phosphorylated a-syn in PD patients than in HCs [31]. Our result demonstrated that the neuronal EV α-syn level was significantly higher in the PD group than in the HC group, and it was positively correlated with the score of the UPDRS part III (Appendix A). The PDD group also exhibited a higher neuronal EV α-syn level than the PD-MCI group, when adjusted for the UPDRS part III score; however, there was no significant difference between these two groups. The results indicate that α-syn is involved in pathological changes in motor symptoms and also the process of cognitive decline. Neuronal EV α-syn might be considered a predictor for the severity of PD. However, we did not see a significant increase in the plasma α-syn level in PD patients compared with HCs. The neuronal EV method might be more sensitive than the plasma method, and the limitations and heterogeneity of our data also lead to this inconsistent result. More specific tests on α-syn, such as 18F-F0502B PET/CT, should be considered in the future, and these may help identify patients at risk for unfavorable disease prognosis and PDD.

Differing from the cortical dementia of AD, PDD is a type of subcortical and cortical dementia which may be related to the location of deposited α-syn [32]. This is why the impaired cognitive domain in PD mainly includes the executive, visuospatial, and attention functions [33]. However, the memory function is thought to be dependent upon the integrity of the Papez circuit, which includes the hippocampus, medial temporal lobe, and cingulate gyrus [34], where the AD-related biomarkers are mainly deposited [35]. About 62% of PD patients showed sufficient pathological manifestations for comorbid AD [36]. A series of studies have indicated a relationship between biomarkers, which showed a relationship between AD and cognitive decline in PD patients [4,5,6,7]. Another study showed that the plasma EV tau level was significantly negatively associated with MMSE and MoCA in PD, and the plasma EV Aβ42 level was not associated with the two batteries mentioned above [11]. In a prospective study [30], lower baseline concentrations of Aβ42 were associated with memory impairment but not with executive–attentional or visuospatial dysfunction [37]. However, the PD patients analyzed in these studies were not divided into different subtypes, which may affect the results regarding the levels of AD-related plasma or neuronal EV biomarkers. Our study showed that the plasma Aβ42 level was significantly lower in the amnestic PD-NC group than in the non-amnestic PD-NC group. Further, the neuronal EV T-tau level was significantly higher in the amnestic PD-MCI group than in the non-amnestic PD-MCI group, which was similar to AD pathology. This demonstrated that in the later period of PD, the pathological proteins of AD are also involved in the process of cognitive decline in PD, which may accelerate the transformation to PDD. These two neurodegenerative diseases will have pathological overlap, although PD is treated as a synuclein disease. This will help clinicians better understand the pathological process of PD. One possible explanation for the lack of a relationship between the level of neuronal EV Aβ in amnestic PD-MCI and non-amnestic PD-MCI could be that the pathogenic Aβ proteins associated with EVs represent only a minor part of the total secreted pathogenic proteins [38,39], and only a small fraction (<1%) of the total Aβ secreted into the extracellular space is shown to be EV-associated [40]. However, we did not find a relationship between AD-related biomarkers in the PDD group, which cannot support the results of the previous postmortem studies [19]. A larger sample in a multicenter follow-up study would be indispensable for exploring the mechanisms of Aβ and tau in cognitive impairment in PD. Further studies in larger PD cognitive cohorts with AD-related PET imaging and CSF are needed to provide more insight.

GFAP, one of the cytoskeletal structure proteins in astrocytes, is highly expressed in the CNS and has been proven to be a useful blood biomarker of astrogliosis in neurodegenerative conditions [40]. In recent years, GFAP has gained attention as a promising biomarker for cognition impairment in PD [41,42,43,44,45]. An increased serum GFAP level was related to a decrease in cognitive scores (total score and multiple cognitive domains) in PD patients [8,46]. Consistent with previous studies, our result demonstrated that the plasma GFAP level was significantly higher in the PD group than in the HC group; however, there were no significant differences in GFAP levels within the different cognitive levels and subtypes in PD-MCI. Longitudinal studies of neuronal EV GFAP in patients with PD are warranted to further validate the correlation of GFAP with cognitive impairment in PD.

A limitation of this study is that it was only a single-center cross-sectional study with a limited sample size, and a prospective large follow-up study is required to confirm our conclusions. As a second limitation, different subtypes of PD-MCI diagnosis were not based on MDS criteria, and we only tested the memory domain for the patients. Other cognitive domain tests should be considered in future studies to identify the different subtypes of PD-MCI. The third limitation is that AD-related CSF biomarkers, including Aβ, T-tau, p-tau181, and GFAP, were not examined in this study for cross comparison; AD-related PET/CT and α-syn-related PET/CT should be considered in future studies. Future studies considering these limitations would be important for verifying our findings and analyses.

## 5. Conclusions

Our study showed that the etiology of cognitive decline in PD is complex, and the impacts of α-syn, tau, and Aβ protein pathology may differ depending on different cognitive domain impairments. The plasma or neuronal EV biomarkers related to AD could be meaningful in explaining the underlying pathological mechanisms of cognitive dysfunction in PD patients and in developing methods of preventing and mitigating this condition.

## Figures and Tables

**Figure 1 brainsci-14-00787-f001:**
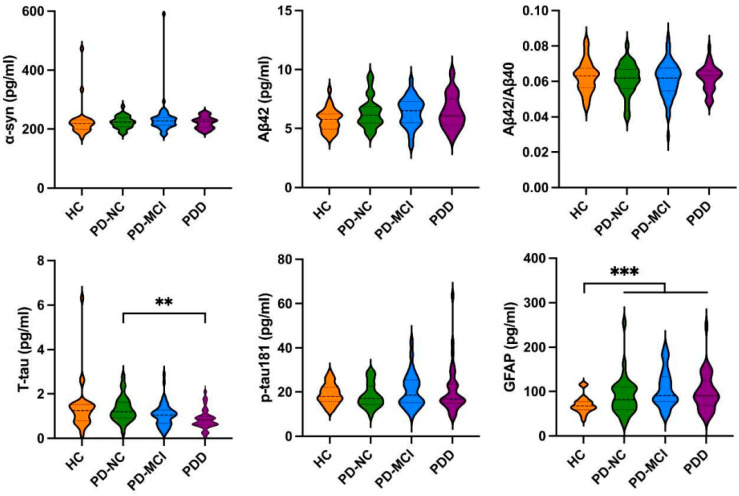
Comparison of plasma biomarkers of α-syn, Aβ42, Aβ42/Aβ40, T-tau, p-tau181 and GFAP levels in HCs, PD-NC, PD-MCI and PDD groups. PD: Parkinson’s disease; HCs: healthy controls; PD-NC: PD with normal cognition; PD-MCI: PD with mild cognitive impairment; PDD: PD with dementia; α-syn: α-synuclein; Aβ: beta-amyloid; T-tau: total tau; p-tau181: phosphorylated tau181; GFAP: glial fibrillary acidic protein. (** *p* < 0.01; *** *p* < 0.001).

**Figure 2 brainsci-14-00787-f002:**
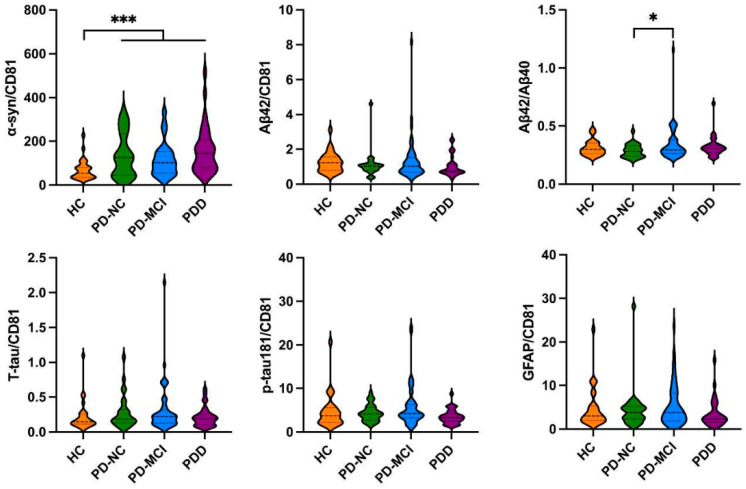
Comparison of neuronal EV biomarkers of α-syn, Aβ42, Aβ42/Aβ40, T-tau, p-tau181 and GFAP levels in HCs, PD-NC, PD-MCI and PDD groups. EV: extracellular vesicle; PD: Parkinson’s disease; HCs: healthy controls; PD-NC: PD with normal cognition; PD-MCI: PD with mild cognitive impairment; PDD: PD with dementia; α-syn: α-synuclein; Aβ: beta-amyloid; T-tau: total tau; p-tau181: phosphorylated tau181; GFAP: glial fibrillary acidic protein; CD81: cluster of differentiation 81. (* *p* < 0.05; *** *p* < 0.001).

**Table 1 brainsci-14-00787-t001:** Clinical characteristics between HC and PD with different cognitive impairment.

	HC (*n* = 30)	PD (*n* = 122)	*χ*2/*p*	PD-NC (*n* = 31)	PD-MCI (*n* = 56)	PDD (*n* = 35)	*χ*2/*p*
Age, years	61.670 ± 5.585	64.010 ± 8.505	0.071	59.390 ± 10.026	64.910 ± 7.655 ^b^	66.660 ± 6.778 ^a,h^	0.001
Sex, male (%)	9 (30.0%)	59 (48.4%)	0.101	10 (32.3%)	33 (58.9%) ^a,b^	16 (45.7%)	0.029
Education, years	13.570 ± 2.555	12.480 ± 2.814	0.055	14.230 ± 1.961	12.640 ± 2.888	10.660 ± 2.248 ^g,h,f^	<0.001
Disease duration, years	NA	5.050 ± 3.836	NA	4.900 ± 3.922	4.690 ± 3.780	5.840 ± 3.865	0.402
Hoehn and Yahr score	NA	3.040 ± 1.366	NA	2.460 ± 1.029	2.790 ± 1.109	4.000 ± 1.581 ^h,i^	<0.001
MDS-UPDRSII score	NA	9.790 ± 7.511	NA	7.310 ± 5.204	9.200 ± 6.350	13.130 ± 9.895 ^b^	0.020
MDS-UPDRSIII score	NA	30.160 ± 15.222	NA	22.440 ± 11.140	28.340 ± 12.430	40.660 ± 17.715 ^h,i^	<0.001
HAMD score	NA	8.190 ± 5.985	NA	6.300 ± 5.664	8.670 ± 5.796	9.160 ± 6.393	0.124
ESS score	NA	4.000 ± 3.622	NA	3.170 ± 2.791	3.960 ± 3.692	4.840 ± 3.348	0.279
Amnestic patient (%)	NA	47 (38.5%)	NA	9 (29.0%)	38(67.9%) ^e^	NA	0.001

MDS-UPDRS II/III: Movement Disorder Society Unified Parkinson’s Disease Rating Scale, part II or III; HAMD: Hamilton Rating Scale for Depression; ESS: Epworth Sleepiness Scale; HC: healthy control; PD: Parkinson’s disease; PD-NC: PD with normal cognition; PD-MCI: PD with mild cognitive impairment; PDD: PD with dementia; NA: not applicable. ^a^ Compared with HC, *p* < 0.05, ^b^ Compared with PD-NC, *p* < 0.05, ^e^ Compared with PD-NC, *p* < 0.01, ^f^ Compared with PD-MCI, *p* < 0.01, ^g^ Compared with HC, *p* < 0.001, ^h^ Compared with PD-NC, *p* < 0.001, ^i^ Compared with PD-MCI, *p* < 0.001.

**Table 2 brainsci-14-00787-t002:** Comparison of biomarker levels between PD-NC or PD-MCI with or without amnestic symptom.

Plasma/Neuronal EV Biomarkers (pg/mL)	Non-Amnestic PD-NC (*n* = 21)	Amnestic PD-NC (*n* = 10)	*p*	Non-Amnestic PD-MCI (*n* = 14)	Amnestic PD-MCI (*n* = 42)	*p*
α-syn	223.997 ± 21.282	222.519 ± 1.366	0.857	219.032 ± 27.335	235.714 ± 58.721	0.327
Aβ42	6.690 ± 1.452	5.579 ± 0.761	0.031	6.360 ± 1.382	6.403 ± 1.208	0.911
Aβ42/Aβ40	0.061 ± 0.010	0.062 ± 0.008	0.770	0.061 ± 0.010	0.061 ± 0.010	0.883
T-tau	1.250 ± 0.628	1.183 ± 0.550	0.775	0.985 ± 0.419	1.038 ± 0.556	0.749
p-tau181	17.941 ± 6.794	17.645 ± 5.519	0.905	18.147 ± 5.855	20.860 ± 8.853	0.289
GFAP	93.653 ± 49.504	84.204 ± 25.117	0.576	91.517 ± 36.880	110.067 ± 42.261	0.576
p-tau181/T-tau	16.852 ± 9.147	15.777 ± 7.767	0.753	23.325 ± 18.282	27.088 ± 19.250	0.654
α-syn/CD81	129.264 ± 106.804	153.352 ± 84.035	0.556	113.011 ± 89.129	118.277 ± 75.006	0.830
Aβ42/CD81	1.151 ± 0.882	1.046 ± 0.283	0.732	1.053 ± 0.450	1.441 ± 1.380	0.310
Aβ42/Aβ40	0.277 ± 0.052	0.304 ± 0.065	0.245	0.339 ± 0.090	0.343 ± 0.163	0.933
T-tau/CD81	0.247 ± 0.176	0.368 ± 0.319	0.194	0.203 ± 0.095	0.337 ± 0.037	0.046
p-tau181/CD81	7.795 ± 4.901	3.960 ± 1.257	0.452	4.803 ± 2.616	5.955 ± 5.132	0.428
GFAP/CD81	18.318 ± 6.680	3.576 ± 1.765	0.498	4.730 ± 3.805	5.638 ± 5.236	0.558
p-tau181/T-tau	38.829 ± 16.747	18.976 ± 13.807	0.389	31.596 ± 16.075	36.075 ± 9.641	0.995

PD: Parkinson’s disease; PD-NC: PD with normal cognition; PD-MCI: PD with mild cognitive impairment; EV: extracellular vesicle; α-syn: α-synuclein; Aβ: beta-amyloid; T-tau: total tau; p-tau181: phosphorylated tau181; GFAP: glial fibrillary acidic protein; CD81: cluster of differentiation 81.

## Data Availability

The raw data are available in a publicly accessible repository, the link to access our raw data is: https://figshare.com/s/1d264eb51875dc8acad8 accessed on 15 July 2024.

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
