# Peer review of "Alzheimer’s Disease Related Biomarkers Were Associated with Amnestic Cognitive Impairment in Parkinson’s Disease: A Cross-Sectional Cohort Study"

_brainsci, 2024, doi:10.3390/brainsci14080787_

Round 1
Reviewer 1 Report
Comments and Suggestions for Authors
In the current study authors have investigated the biomarkers of PD and AD in plasma and Neuronal exosomes. Authors found that there is a change in GFAP levels in plasma between the disease and healthy controls and Alpha-synuclein can be a sensitive biomarker for assisting diagnostic and predicting disease severity of PD. Study is well designed written and I really enjoyed reading it. I have several small comments, suggestions and questions which may further improve the manuscript.
1. In the measurement of p-181 Tau and Total-Tau, also calculate the p-181 Tau/Total-Tau, which may a meaningful difference.
2. Where there is a meaningful difference, plot a ROC curve.
3. Please use the term extracellular vesicles instead of exosomes as per the ISEV guidelines.
4. Check the neuronal markers in the EVs isolated using the kit to make sure they are enriched neuronal derived.
Author Response
Dear Editor, Dear reviewers
Thank you for your letter dated June 24. We were pleased to know that our work was rated as potentially acceptable for publication in Journal, subject to adequate revision. We thank the reviewers for the time and effort that they have put into reviewing the previous version of the manuscript. Their suggestions have enabled us to improve our work.
We uploaded the revised manuscript and our responses are given directly afterward in a different color (red).
We hope that the revised manuscript is accepted for publication in the Journal of Brain Sciences.
Sincerely, Xiaofan Xue.
Comment 1: In the measurement of p-181 Tau and Total-Tau, also calculate the p-181 Tau/Total-Tau, which may a meaningful difference.
|
p-tau181/T-tau |
PD |
HC |
p |
|
Neuronal EVs |
28.601±12.725 |
26.770±19.944 |
0.829 |
|
plasma |
32.304±13.283 |
27.276±15.685 |
0.775 |
Response1: Thank you for this important suggestion. We have calculated the plasma and NDE(which I mentioned as Neuronal extracellular vesicle (EV) below) p-181 Tau/Total-Tau ration between different groups. The results are below:
|
|
PD-NC |
PD-MCI |
PDD |
HC |
p |
|
NDE |
32.668±16.259 |
31.663±16.743 |
20.386±10.853 |
26.770±19.944 |
0.556 |
|
plasma |
16.494±8.591 |
26.130±16.776 |
55.558±18.570 |
27.276±15.685 |
0.068 |
|
p-tau181/T-tau |
Non-amnestic PD-NC |
Amnestic PD-NC |
p |
|
Neuronal EVs |
38.829±16.747 |
18.976±13.807 |
0.389 |
|
plasma |
16.852±9.147 |
15.777±7.767 |
0.753 |
|
p-tau181/T-tau |
Non-amnestic PD-MCI |
Amnestic PD-MCI |
p |
|
Neuronal EVs |
31.596±16.075 |
36.075±9.641 |
0.995 |
|
plasma |
23.325±18.282 |
27.088±19.250 |
0.654 |
We have added this result into the table of Table 2 (Mentioned exactly where in the revised manuscript this change can be found-page7and Supplementary Table 1.)
Comment 2: Where there is a meaningful difference, plot a ROC curve.
Response 2: Thank you for this important suggestion. We have added the ROC curve for four meaningful biomarkers. (Supplementary Figure 2 and Supplementary Table 3-4) (Mentioned exactly where in the revised manuscript this change can be found-page9-10, line359-364, line367-369)
Figure 2 (a-b) ROC curve for discriminating the PD group from the HC group using plasma GFAP level and NDE α-syn values as diagnostic parameters. (c) ROC curve for discriminating the Non-amnestic PD-NC group from the amnestic group using plasma Aβ42 level value as diagnostic parameters. (d) ROC curve for discriminating the Non-amnestic PD-MCI group from the amnestic group using NDE T-tau/CD81 level value as diagnostic parameters. Abbreviations: AUC, area under the receiver operating characteristic curve.
Supplementary Table 3. ROC analyses of plasma Aβ42 level for predicting of amnestic PD-NC versus non-amnestic PD-NC
|
Plasma Biomarkers Non-amnestic PD-NC VS amnestic PD-NC |
Cutoff value |
Sensitivity; % |
Specificity; % |
AUC (95% CI) |
P value |
|
Aβ42 |
6.510 pg/ml |
57.1 |
90.0 |
0.6905 (0.5046-0.8763) |
0.091 |
Abbreviations: PD: Parkinson’s Disease; PD-NC: PD with normal cognition; AUC, area under the receiver operating characteristic curve; CI, confidence interval.
Supplementary Table 4. ROC analyses of NDE T-tau level for predicting of amnestic PD-MCI versus non-amnestic PD-MCI
|
Neuronal EV biomarkers Non-amnestic PD-MCI VS amnestic PD-MCI |
Cutoff value |
Sensitivity; % |
Specificity; % |
AUC (95% CI) |
P value |
|
T-tau/CD81 |
0.315 |
30.8 |
92.9 |
0.5714 (0.4156-0.7272) |
0.431 |
Abbreviations: PD: Parkinson’s Disease; PD-MCI: PD with mild cognitive impairment; AUC, area under the receiver operating characteristic curve; CI, confidence interval.
Comment 3: Please use the term extracellular vesicles instead of exosomes as per the ISEV guidelines.
Response 3: Thank you for this important suggestion. We repalced all the Neuronal Derived Exosome (NDE) as neuronal extracellular vesicles (EVs).
Comment 4: Check the neuronal markers in the EVs isolated using the kit to make sure they are enriched neuronal derived
Response 4: Thank you for this important suggestion. Actually, all the biomarkers (plasma and Neuronal EVs) were measured using Single-molecule array and Chemiluminescence immunoassay respectively, not using the Western blotting. WB analysis was performed to verify the presence of typical exosome markers in Neuronal EVs. So we revised the sentence in abtract and also in the 2.4 part of Materials and Methods. By the way, we offer another supplementary figure about the details about the presence of typical EV markers in Neuronal EVs by WB. (Mentioned exactly where in the revised manuscript this change can be found-page1, line31; page4, line151-163; page9, line355-359)

Reviewer 2 Report
Comments and Suggestions for Authors
Review of the manuscript “Plasma Aβ42 and Neuronal Derived Exosome T-tau were associated with Amnestic Cognitive Impairment in Parkinson’s Disease” by Xiaofan Xue and coauthors.
Cognitive dysfunction frequently occurs as a non-motor symptom in Parkinson’s disease and is linked to diminished quality of life. Research indicates that α-synuclein toxicity in PD can lead to neuronal cell death and impairment in cognitive function. The authors examined whether neuronal-derived exosome levels of α-synuclein can be used as a biomarker for diagnosis and predicting Parkinson’s disease severity. This is an important biomedical area and the results presented in the manuscript will be interesting for the readers of the “Brain Sciences”
The following corrections should be made.
Abstract.
-“Background: Cognitive impairment is common in patients with Parkinson’s disease (PD) and pathologies of both Alzheimer’s disease (AD) and α-synucleopathies were the potential contributing mechanisms.” The sense of this sentence is unclear making the text not focused and hard to follow.
-“122 PD patients without and with MCI and dementia” Abbreviations may be used only after giving the full name of the term: mild cognitive impairment (MC).
-There is a contradiction between the Title focusing on Parkinson’s disease and the Abstract, and other parts of the manuscript dedicated to both Parkinson’s and Alzheimer’s disease.
Materials and Methods:
-“Chemiluminescence immunoassay and Western Blotting respectively” All details of Western blotting should be given, including antibodies used, their dilutions, condition of PAGE, transfer to membrane, type of membrane, etc.
- The diagnosis of PD-MCI patients exclusively based on memory domain is not sufficient. It should include MDS criteria and other cognitive domains.
Results
-The disadvantage of the manuscript is that the authors do not give the real experimental results, such as figure of the western blots, but show only data after calculations which are not very convincing.
Discussion
-“Pathologically, PD is characterized by degeneration of the dopaminergic nigrostriatal system including the over deposition of α-syn in Lewy bodies (LBs) and Lewy neurites.” After this sentence the aauthors should add the following reference :”Emamzadeh et al. Parkinson’s disease: Biomarkers, Treatment, and Risk Factors. Frontiers in Neuroscience, Neurodegeneration, 12, 61230, 2018. https://doi.org/10.3389/fnins.2018.00612
-The authors should explain their opinion why plasma Aβ42 and neuronal derived exosome T-tau, but not typical Parkinson’s disease markers are associated with amnestic cognitive impairment in Parkinson’s Disease.
Author Response
Dear Editor, Dear reviewers
Thank you for your letter dated June 18. We were pleased to know that our work was rated as potentially acceptable for publication in Journal, subject to adequate revision. We thank the reviewers for the time and effort that they have put into reviewing the previous version of the manuscript. Their suggestions have enabled us to improve our work.
Please forgive us cannot resubmit a revised copy of the manuscript immediately, because our manuscript need to have English Editing which takes more time. We will uploaded the revised manuscript as soon as possible and our responses are given directly afterward in a different color (red).
We hope that the revised manuscript is accepted for publication in the Journal of Brain Sciences.
Sincerely, Xiaofan Xue.
Comment 1: “Background: Cognitive impairment is common in patients with Parkinson’s disease (PD) and pathologies of both Alzheimer’s disease (AD) and α-synucleopathies were the potential contributing mechanisms.” The sense of this sentence is unclear making the text not focused and hard to follow.
Response 1: Thank you for pointing this out. We agree with this comment. Therefore, we have replaced this sencentence with “Cognitive impairment is common in patients with Parkinson’s disease (PD) and occurs through multiple mechanisms, including Alzheimer’s disease (AD) pathology and the involvement of α-synucleinopathies” (Mentioned exactly where in the revised manuscript this change can be found-page1, line 21-23)
Comment 2: “122 PD patients without and with MCI and dementia” Abbreviations may be used only after giving the full name of the term: mild cognitive impairment (MC).
Response 2: Thank you for pointing this out. We agree with this comment. Therefore, we have revised this part already.
Comment 3: There is a contradiction between the Title focusing on Parkinson’s disease and the Abstract, and other parts of the manuscript dedicated to both Parkinson’s and Alzheimer’s disease.
Response 3: Thank you for pointing this out. Actually speaking, there is no contradiction between the title and abstract. We mentioned “AD pathological biomarkers” in abstract which include the “plamsa Aβ42 and NDE(which I mentioned as Neuronal extracellular vesicles (EVs) below) T-tau”. Or, we can revise the title with “AD related biomarkers were associated with Amnestic Cognitive Impairment in Parkinson’s Disease: A Cross-sectional Cohort Study”.(Mentioned exactly where in the revised manuscript this change can be found-page1, line2-4)
Comment 4: “Chemiluminescence immunoassay and Western Blotting respectively” All details of Western blotting should be given, including antibodies used, their dilutions, condition of PAGE, transfer to membrane, type of membrane, etc.
Response 4: Thank you for pointing this out. We have discussed with the researchers who underwent the testing about the biomarkers and all the biomarkers (plasma and Neuronal EVs were measured using Single-molecule array and Chemiluminescence immunoassay respectively, not using the Western blotting. Actually, WB analysis was performed to verify the presence of typical exosome markers in Neuronal EVs. So we revised the sentence in abtract and also in the 2.4 part of Materials and Methods. (Mentioned exactly where in the revised manuscript this change can be found-page1, line31; page4, line151-163)
Comment 5: The diagnosis of PD-MCI patients exclusively based on memory domain is not sufficient. It should include MDS criteria and other cognitive domains.
Response 5: Thank you for pointing this out. Actually, there are two levels of the diagnostic of PD-MCI. The level I allows for the diagnostic of PD-MCI based on an abbreviated cognitive assessment, such as MoCA. However, for the diagnosis of the different subtypes of PD-MCI, the task force recommands formal, comprehensive neuropsychological testing that includes at least two tests for each cognitive domain. If one or more than one tests performance between 1 to 2 SD significant decline from estimated premobid levels, it indicates that the patient get impairment on that cognitive domain. (Reference 16) For this study, we only focused on the amnestic and non-amnestic PD-MCI, that’s why we assessed 2 memory cognitive domain tests for all the participants. However, we mentioned about that other cognitive domains test should be considered in further study for identify the different subtypes of PD-MCI, not only the amnestic subtype. (Mentioned exactly where in the revised manuscript this change can be found-page8-9, line 339-342)
Comment 6: The disadvantage of the manuscript is that the authors do not give the real experimental results, such as figure of the western blots, but show only data after calculations which are not very convincing.
Response 6: Thank you for pointing this out. Same as comment 4, we revised the sentence in abtract and also in the 2.4 part of Materials and Methods. By the way, we offer another supplementary figure about the details about the presence of typical exosome markers in Neuronal EVs by WB. (Mentioned exactly where in the revised manuscript this change can be found-page1, line31; page4, line151-163)
Comment 7:“Pathologically, PD is characterized by degeneration of the dopaminergic nigrostriatal system including the over deposition of α-syn in Lewy bodies (LBs) and Lewy neurites.” After this sentence the aauthors should add the following reference :”Emamzadeh et al. Parkinson’s disease: Biomarkers, Treatment, and Risk Factors. Frontiers in Neuroscience, Neurodegeneration, 12, 61230, 2018. https://doi.org/10.3389/fnins.2018.00612
Response 7: Thank you for the suggestion, we would add this reference after the sentence above.(Reference 29)
Comment 8: The authors should explain their opinion why plasma Aβ42 and neuronal derived exosome T-tau, but not typical Parkinson’s disease markers are associated with amnestic cognitive impairment in Parkinson’s Disease.
Response8: Thank you for this suggestion. Firstly, this is a single-center cross-sectional cohort study with limited sample size and we only observed the level of different biomarkers with PD-CI. We tried to give a cautious overall interpretation of the result but not a definitive explanation. Differing from the cortical dementia of AD, PDD is type of subcortical and cortical dementia which may be related to the location of deposited α-syn. This is why the impaired cognitive domain in PD mainly includes the executive, visuospatial, and attention functions. Our result demonstrated that the Neuronal EV α-syn level was significantly higher in the PD group than HC group, and it was positively correlated with the score of UPDRS part III (Supplementary Table 4). PDD also exhibited higher Neuronal EV α-syn level than did PD-MCI group adjusted for the score of UPDRS part III. It indicated that α-syn maybe involved in the pathological changes of motor symptoms and also the process of cognitive decline, but not the amnestic subtype of PD-MCI. As we all know, the memory function is thought to be depended upon the integrity of the Papez circuit, which includes the hippocampus, parahippocampal gyrus, medial temporal lobe, mammillary bodies, insula, and cingulate gyrus. And Aβ protein mainly deposited in the anterior cuneus and posterior cingulate gyrus, and Tau protein mainly gathered in the medial temporal lobe. We hypothesized that there maybe a relationship between the bimarkers of AD and amnestic subtype of PD-MCI. Interestingly, we found that plasma Aβ42 and Neuronal EV T-tau had a significantly relationship with amnestic PD-MCI, but not typical PD biomarkers. However, we did not found the same conclusion with the classic AD biomarkers, such as Aβ42/40 and p-tau181. We tried to describe the reason why these happened in the discussion part, and future studies would be important for verifying our findings.(Mentioned exactly where in the revised manuscript this change can be found-page8, line292-298)

Reviewer 3 Report
Comments and Suggestions for Authors
Few comments:
1. This spectrum of disease has a sexually dimorphic and age related components association. What is the concentration of the plasma biomarkers among males and females; and at different sub-age groups (show data as figures)? If you do not see any difference, how do you explain that observation with clinical relevance?
2. How this research will advance the understanding of the neurodegenerative disease field? Please add a section discussing that.
Author Response
Dear Editor, Dear reviewers
Thank you for your letter dated June 20. We were pleased to know that our work was rated as potentially acceptable for publication in Journal, subject to adequate revision. We thank the reviewers for the time and effort that they have put into reviewing the previous version of the manuscript. Their suggestions have enabled us to improve our work.
We uploaded the revised manuscript and our responses are given directly afterward in a different color (red).
We hope that the revised manuscript is accepted for publication in the Journal of Brain Sciences.
Sincerely, Xiaofan Xue.
Comment 1: “Background: Cognitive Comment 1: This spectrum of disease has a sexually dimorphic and age related components association. What is the concentration of the plasma biomarkers among males and females; and at different sub-age groups (show data as figures)? If you do not see any difference, how do you explain that observation with clinical relevance?
Response1: Thank you for pointing it out. Auctually we showed the data in the part of “3.4 Correlation between clinical characteristic and levels of biomarkers”. As you said, this spectrum of disease has a sexually dimorphic and age related components association, “It was found that age was negatively correlated with plasma levels Aβ42/Aβ40 ratio, but positively with levels of GFAP. NDE(which I mentioned as Neuronal extracellular vesicle (EV) below) T-tau level was also positively with the percentage of male” (Supplementary Table 3-4). The average age of the participants we incruited is about PD: 64.010±8.505 years, HC: 61.670±5.585 years. That’s why we didn’t calculate the data of sub-age groups because the limited range.
Comment 2: How this research will advance the understanding of the neuro- degenerative disease field? Please add a section discussing that.
Response2: Thank you for this suggestion. We have added several sentences for discussing this issue in the new version of manuscript. Differing from the dementia of AD, PDD is type of subcortical and cortical dementia which may be related to the location of deposited α-syn. This is why the impaired cognitive domain in PD mainly includes the executive, visuospatial, and attention functions. Our result demonstrated that α-syn maybe involved in the pathological changes of motor symptoms and also the process of cognitive decline, but not the amnestic subtype of PD-MCI. Interestingly, we found that plasma Aβ42 and neuronal EV T-tau had a significantly relationship with amnestic PD-MCI, but not typical PD markers. The results above maybe demonstrate that the later period of PD, the pathological proteins of AD are also involved in the process of PD cognitive decline, which may accelerate the transformation to PDD. These two neurodegenerated diseases will have pathological overlap, although PD is treated as a synuclein disease. This will help clinician to understand the pathological process of PD better. (Mentioned exactly where in the revised manuscript this change can be found-page8, line292-298; page9, 311-316)

Reviewer 4 Report
Comments and Suggestions for Authors
Firstly, I would like to thank the authors for taking the time to submit their work to our Journal. I have a few suggestions:
1) The rate of plagiarism at iThenticate report is considerably high. Please modify your text so that the plagiarism rates decrease.
2) We suggest summarizing and decreasing the word count of the title. However, please add the study design type to the title.
3) Please refrain from using the term “demented” which could be negatively interpreted by patients.
4) We advise using English editing services to improve the readability of your paper.
5) Please include the study design type and the period of recruitment in the abstract.
6) Please add epidemiological data in percentages in the introduction
7) L61-62: Please add reference to this sentence
8) Please review the text regarding references, they seem to be missing in the end of several sentences.
9) In the last paragraph of the introduction, authors should clearly state their study hypothesis and their research question. The primary and secondary goals of the study considering the previous literature should also be detailed.
10) Overall, in the introduction the authors must try to explain what is new and original about their article.
11) Was your study protocol previously validated?
12) What was the frequency in which the patients were seen and by which professionals they were assessed?
13) Please explain why the authors chose to assess these particular biomarkers
14) Detail inclusion and exclusion criteria and why they were chosen
15) How did the authors arrive to the n of the study?
16) Was the power of the study calculated?
17) Please detail the study design type in the methods.
18) We recommend following the STROBE guidelines for this study
19) Was normality assessed in this study?
20) Please explain rationale behind each of the questionnaires used , how they were chosen and validated to your population, and if no copyrights were violated in this study
21) Please provide a detailed table with baseline characteristics of the patients, including comorbidities, ethinicity, social history, medications in use and dosage, time ssince diagnosis, presenting symptoms
22) Authors should expand their limitations section and be careful with the interpretation of their results. It is unlikely that predictions can be made in a study of this type.
23) Please explain point by point to each of the STROBE questionnaire questions
|
24) |
Item No |
Recommendation |
|
Title and abstract |
1 |
(a) Indicate the study’s design with a commonly used term in the title or the abstract |
|
(b) Provide in the abstract an informative and balanced summary of what was done and what was found |
||
|
Introduction |
||
|
Background/rationale |
2 |
Explain the scientific background and rationale for the investigation being reported |
|
Objectives |
3 |
State specific objectives, including any prespecified hypotheses |
|
Methods |
||
|
Study design |
4 |
Present key elements of study design early in the paper |
|
Setting |
5 |
Describe the setting, locations, and relevant dates, including periods of recruitment, exposure, follow-up, and data collection |
|
Participants |
6 |
(a) Cohort study—Give the eligibility criteria, and the sources and methods of selection of participants. Describe methods of follow-up Case-control study—Give the eligibility criteria, and the sources and methods of case ascertainment and control selection. Give the rationale for the choice of cases and controls Cross-sectional study—Give the eligibility criteria, and the sources and methods of selection of participants |
|
(b) Cohort study—For matched studies, give matching criteria and number of exposed and unexposed Case-control study—For matched studies, give matching criteria and the number of controls per case |
||
|
Variables |
7 |
Clearly define all outcomes, exposures, predictors, potential confounders, and effect modifiers. Give diagnostic criteria, if applicable |
|
Data sources/ measurement |
8* |
For each variable of interest, give sources of data and details of methods of assessment (measurement). Describe comparability of assessment methods if there is more than one group |
|
Bias |
9 |
Describe any efforts to address potential sources of bias |
|
Study size |
10 |
Explain how the study size was arrived at |
|
Quantitative variables |
11 |
Explain how quantitative variables were handled in the analyses. If applicable, describe which groupings were chosen and why |
|
Statistical methods |
12 |
(a) Describe all statistical methods, including those used to control for confounding |
|
(b) Describe any methods used to examine subgroups and interactions |
||
|
(c) Explain how missing data were addressed |
||
|
(d) Cohort study—If applicable, explain how loss to follow-up was addressed Case-control study—If applicable, explain how matching of cases and controls was addressed Cross-sectional study—If applicable, describe analytical methods taking account of sampling strategy |
||
|
(e) Describe any sensitivity analyses |
||
Continued on next page
|
Results |
||
|
Participants |
13* |
(a) Report numbers of individuals at each stage of study—eg numbers potentially eligible, examined for eligibility, confirmed eligible, included in the study, completing follow-up, and analysed |
|
(b) Give reasons for non-participation at each stage |
||
|
(c) Consider use of a flow diagram |
||
|
Descriptive data |
14* |
(a) Give characteristics of study participants (eg demographic, clinical, social) and information on exposures and potential confounders |
|
(b) Indicate number of participants with missing data for each variable of interest |
||
|
(c) Cohort study—Summarise follow-up time (eg, average and total amount) |
||
|
Outcome data |
15* |
Cohort study—Report numbers of outcome events or summary measures over time |
|
Case-control study—Report numbers in each exposure category, or summary measures of exposure |
||
|
Cross-sectional study—Report numbers of outcome events or summary measures |
||
|
Main results |
16 |
(a) Give unadjusted estimates and, if applicable, confounder-adjusted estimates and their precision (eg, 95% confidence interval). Make clear which confounders were adjusted for and why they were included |
|
(b) Report category boundaries when continuous variables were categorized |
||
|
(c) If relevant, consider translating estimates of relative risk into absolute risk for a meaningful time period |
||
|
Other analyses |
17 |
Report other analyses done—eg analyses of subgroups and interactions, and sensitivity analyses |
|
Discussion |
||
|
Key results |
18 |
Summarise key results with reference to study objectives |
|
Limitations |
19 |
Discuss limitations of the study, taking into account sources of potential bias or imprecision. Discuss both direction and magnitude of any potential bias |
|
Interpretation |
20 |
Give a cautious overall interpretation of results considering objectives, limitations, multiplicity of analyses, results from similar studies, and other relevant evidence |
|
Generalisability |
21 |
Discuss the generalisability (external validity) of the study results |
|
Other information |
||
|
Funding |
22 |
Give the source of funding and the role of the funders for the present study and, if applicable, for the original study on which the present article is based |
Comments on the Quality of English Language
Extensive language editing required
Author Response
Dear Editor, Dear reviewers
Thank you for your letter dated June 22. We were pleased to know that our work was rated as potentially acceptable for publication in Journal, subject to adequate revision. We thank the reviewers for the time and effort that they have put into reviewing the previous version of the manuscript. Their suggestions have enabled us to improve our work.
We uploaded the revised manuscript and our responses are given directly afterward in a different color (red).
We hope that the revised manuscript is accepted for publication in the Journal of Brain Sciences.
Sincerely, Xiaofan Xue.
Comment 1: The rate of plagiarism at iThenticate report is considerably high. Please modify your text so that the plagiarism rates decrease.
Response 1: Thank you for pointing it out, we have revised the whole manuscript and had the English editing already to decrease the plagiarism rates.
Comment 2: We suggest summarizing and decreasing the word count of the title. However, please add the study design type to the title.
Response 2: Thank you for the suggestion, we revised out title as “AD related biomarkers were associated with Amnestic Cognitive Impairment in Parkinson’s Disease: a cross-sectional cohort study”. (Mentioned exactly where in the revised manuscript this change can be found-page1, line2-4)
Comment 3: Please refrain from using the term “demented” which could be negatively interpreted by patients.
Response 3: Thank you for pointing it out, we have replaced the “demented” with “dementia”. (Mentioned exactly where in the revised manuscript this change can be found-page1,line27)
Comment 4: We advise using English editing services to improve the readability of your paper.
Response 4: Thank you for the suggestion. We have improved the readability with the English editing on MDPI website.
Comment 5: Please include the study design type and the period of recruitment in the abstract.
Response 5: Thank you for the suggestion. We have invised the abstract as “A total of 122 patients with PD and 30 healthy controls were included in this cross-sectional cohort study between March 2021 and July 2023”. Mentioned exactly where in the revised manuscript this change can be found-page1, line25-27)
Comment 6: Please add epidemiological data in percentages in the introduction
Response 6: Thank you for the suggestion. We have revised the Introduction already with epidemiological data in percentages. (Mentioned exactly where in the revised manuscript this change can be found-page1, line45; page2, line49)
Comment 7: L61-62: Please add reference to this sentence
Response 7: We have add the reference [14] after the sentence at L66-67.
Comment 8: Please review the text regarding references, they seem to be missing in the end of several sentences.
Response 8: Thank you for pointing it out. We have reviewed all the text and have the 40 references updated to a total of 47 references.
Comment 9: In the last paragraph of the introduction, authors should clearly state their study hypothesis and their research question. The primary and secondary goals of the study considering the previous literature should also be detailed.
Response 9: Thank you for the suggestion. Actually, we deleted a part of the last paragraph of the introduction to decrease the plagiarism rates. We rewrote the last paragraph again with the primary and secondary goals of the study. (Mentioned exactly where in the revised manuscript this change can be found-page2, line85-91)
Comment 10: Overall, in the introduction the authors must try to explain what is new and original about their article.
Response 10: Thank you for the suggestion. We rewrote the last paragraph with the new and original about their article. (Mentioned exactly where in the revised manuscript this change can be found-page2, line87-93)
Comment 11: Was your study protocol previously validated?
Response 11:Thank you for pointing this out. Yes, all the testing protocol were validated and well applicated in the previous study.
Comment 12: What was the frequency in which the patients were seen and by which professionals they were assessed?
Response 12: Most of PD patients were examined regularly at Xuanwu Hospital, and the frequency of visits ranged from one to several times. All the clinical and neuropsychological assessments were underwent by professional and experienced neurologists.
Comment 13: Please explain why the authors chose to assess these particular biomarkers
Response 13: Thank you for pointing this out. . Differing from the cortical dementia of AD, PDD is type of subcortical and cortical dementia which may be related to the location of deposited α-syn. This is why the impaired cognitive domain in PD mainly includes the executive, visuospatial, and attention functions. However, the memory dysfunction is always related with the Papez circuit, which includes the hippocampus, parahippocampal gyrus, medial temporal lobe, mammillary bodies, insula, and cingulate gyrus. Previous data showed the pathological proteins of AD are involved in the process of PD cognitive decline, which may accelerate the transformation to PDD. These two neurodegenerated diseases will have pathological overlap. As we all know, Aβ protein mainly deposite in the anterior cuneus and posterior cingulate gyrus, and Tau protein mainly gathered in the medial temporal lobe of AD. We hypothesized that there maybe a relationship between the bimarkers of AD and amnestic subtype of PD-MCI.That’s why we chose these biomarkers in this study. (Mentioned exactly where in the revised manuscript this change can be found-page8, line292-298)
Comment 14: Detail inclusion and exclusion criteria and why they were chosen
Response 14: All patients were diagnosed as clinical or probable PD according to the Movement Disorder Society (MDS) Diagnostic Criteria for Parkinson’s Disease (Reference 20) by movement disorder specialists. The exclusion criteria were updated with (1) History of serious diseases affecting brain function: cerebral vascular diseases (Fazakes≥II; Hachinski>4), cerebral trauma, encephalitis, epilepsy et al; (2) History of other serious diseases affecting cognitive function: severe mental diseases, cancer, hyperthyroidism, carbon monoxide poisoning, intemperance, drug addiction, and severe systemic diseases; (3) Poor cooperation: auditory and visual impairment, aphasia and severe paralysis; (4) History of taking anticholinergic drugs. All the conditions are the potential reason for cognitive impairment. (Mentioned exactly where in the revised manuscript this change can be found-page3, line102-108)
Comment 15: How did the authors arrive to the n of the study?
Response 15: Thank you for pointing this out. This is a single center of clinical cross-section cohort study. Combined with the relevant research on PD cognitive scale in previous literature, the preliminary sample size is 120 cases, another 30 cases of normal control group were included.
Comment 16: Was the power of the study calculated?
Response 16: Thank you for this comment. Same as comment 15. We did calculated the power of this study.
Comment 17: Please detail the study design type in the methods.
Response 17: Thank you for the suggestion. We have invised the 2.1 as “In this cross-sectional cohort study, we recruited 122 consecutive patients with PD admitted to the Clinical and Research Center of Xuanwu Hospital of Capital Medical University in Beijing, China between March 2021 and July 2023 ”. (Mentioned exactly where in the revised manuscript this change can be found-page 3, line102-108)
Comment 18: We recommend following the STROBE guidelines for this study
Response 18: Thank you for the recommend. We would follow the STROBE guidelines for this study.
Comment 19: Was normality assessed in this study?
Response 19: Yes, we recruited 30 healthy controls as the normal group.
Comment 20: Please explain rationale behind each of the questionnaires used , how they were chosen and validated to your population, and if no copyrights were violated in this study
Response 20: Thank you for pointing it out. Actually, all the clinical or neuropsychological assessments are widly used and open access. Meanwhile, all the cognitive assessments were recommended by Diagnostic criteria for mild cognitive impairment in Parkinson’s disease. There is no copyrights were violated in this study.
Comment 21: Please provide a detailed table with baseline characteristics of the patients, including comorbidities, ethinicity, social history, medications in use and dosage, time since diagnosis, presenting symptoms
Response 21: Thank you for the suggestion. Actually speaking, the PD patients we recruited under strictly screening by the inclusion and exclusion criteria in this study, and the comorbidities and social history are clean. 98% of the participants are Han Chinese. We have the “Disease duration” as the time since diagnosis. We have Hoehn & Yahr and UPDRS-III to represent the motor symptoms and UPDRS-II as the activities of daily living in Table 1.
Comment 22: Authors should expand their limitations section and be careful with the interpretation of their results. It is unlikely that predictions can be made in a study of this type.
Response 22: Thank you for this suggestion. This is a single-center cross-sectional study with limited sample size and we only observed the level of different biomarkers with several groups. We tried to give a cautious overall interpretation of the result but not a definitive explanation or prediction. Under you considerable suggestion, we have revised the discussion of the manuscript. (Mentioned exactly where in the revised manuscript this change can be found-page8, line292-298; page9, 311-316)
Comment 23: Please explain point by point to each of the STROBE questionnaire questions.
Response 1a: Thank you for the comment. We have revised the title and abstract with the study’s design. (Mentioned exactly where in the revised manuscript this change can be found-page1, line2-4, line25-26)
Response 1b: Thank you for the comment. We have provided an informative and balanced summary of what was done and what was found.
Response 2: We have explained the scientific background for the investigation being reported in the Introduction.
Response 3:Thank you for the comment. We have stated specific objectives, including any prespecified hypotheses in the last paragraph of Introduction. (Mentioned exactly where in the revised manuscript this change can be found-page2, line87-93)
Response 4: We have presented key elements of study design.
Response 5: We have describe the period of recruitment and data collection. (Mentioned exactly where in the revised manuscript this change can be found-page3, line97-99)
Response 6a: Thank you for the comment. It is a cross-sectional cohort study and we have modified the eligibility criteria, and the sources and methods of selection of participants. (Mentioned exactly where in the revised manuscript this change can be found-page3, line102-108)
Response 7: This study is not applicable.
Response 8: Thank you for the comment. We have described the detailed Statistical analysis of 2.5. (Mentioned exactly where in the revised manuscript this change can be found-page4, line169-180)
Response 9: To address potential sources od bias, we tried to have clear inclusion and exclusion criteria, keep the consistency of the evaluator's criteria and the time of sample collection. All the plasma and NDE(which I mentioned as Neuronal extracellular vesicle (EV) below) samples were tested in duplicate. (page3, line102-108, line137)
Response 10: Thank you for the comment. This is a single center of clinical cross-sectional cohort study. Combined with the relevant research on PD cognitive scale in previous literature, the preliminary sample size is 120 cases, another 30 cases of normal control group were included.
Response 11: This study is not applicable.
Response 12: Thank you for the comment. We have described the statistical methods in the manuscript. (Mentioned exactly where in the revised manuscript this change can be found-page4, line169-180)
Response 13-17: Thank you for the comment. We have described the details about the participants, the clinical information of study participants, the outcome events and the subgroups analyses in Table1 and Supplementary Table 2.
Response 18-21: Thank you for the comment. We have summarized key results, limitation of this study, a cautious overall interpretation of results and the generalisability of the study results in the part of discussion. (Mentioned exactly where in the revised manuscript this change can be found-page7-9, line256-346)
Response 22: This study is not applicable.
|
|
Item No |
Recommendation |
|
Title and abstract |
1 |
(a) Indicate the study’s design with a commonly used term in the title or the abstract |
|
(b) Provide in the abstract an informative and balanced summary of what was done and what was found |
||
|
Introduction |
||
|
Background/rationale |
2 |
Explain the scientific background and rationale for the investigation being reported |
|
Objectives |
3 |
State specific objectives, including any prespecified hypotheses |
|
Methods |
||
|
Study design |
4 |
Present key elements of study design early in the paper |
|
Setting |
5 |
Describe the setting, locations, and relevant dates, including periods of recruitment, exposure, follow-up, and data collection |
|
Participants |
6 |
(a) Cohort study—Give the eligibility criteria, and the sources and methods of selection of participants. Describe methods of follow-up Case-control study—Give the eligibility criteria, and the sources and methods of case ascertainment and control selection. Give the rationale for the choice of cases and controls Cross-sectional study—Give the eligibility criteria, and the sources and methods of selection of participants |
|
(b) Cohort study—For matched studies, give matching criteria and number of exposed and unexposed Case-control study—For matched studies, give matching criteria and the number of controls per case |
||
|
Variables |
7 |
Clearly define all outcomes, exposures, predictors, potential confounders, and effect modifiers. Give diagnostic criteria, if applicable |
|
Data sources/measurement |
8* |
For each variable of interest, give sources of data and details of methods of assessment (measurement). Describe comparability of assessment methods if there is more than one group |
|
Bias |
9 |
Describe any efforts to address potential sources of bias |
|
Study size |
10 |
Explain how the study size was arrived at |
|
Quantitativevariables |
11 |
Explain how quantitative variables were handled in the analyses. If applicable, describe which groupings were chosen and why |
|
Statisticalmethods |
12 |
(a) Describe all statistical methods, including those used to control for confounding |
|
(b) Describe any methods used to examine subgroups and interactions |
||
|
(c) Explain how missing data were addressed |
||
|
(d) Cohort study—If applicable, explain how loss to follow-up was addressed Case-control study—If applicable, explain how matching of cases and controls was addressed Cross-sectional study—If applicable, describe analytical methods taking account of sampling strategy |
||
|
(e) Describe any sensitivity analyses |
||
Continued on next page
|
Results |
||
|
Participants |
13* |
(a) Report numbers of individuals at each stage of study—eg numbers potentially eligible, examined for eligibility, confirmed eligible, included in the study, completing follow-up, and analysed |
|
(b) Give reasons for non-participation at each stage |
||
|
(c) Consider use of a flow diagram |
||
|
Descriptive data |
14* |
(a) Give characteristics of study participants (eg demographic, clinical, social) and information on exposures and potential confounders |
|
(b) Indicate number of participants with missing data for each variable of interest |
||
|
(c) Cohort study—Summarise follow-up time (eg, average and total amount) |
||
|
Outcome data |
15* |
Cohort study—Report numbers of outcome events or summary measures over time |
|
Case-control study—Report numbers in each exposure category, or summary measures of exposure |
||
|
Cross-sectional study—Report numbers of outcome events or summary measures |
||
|
Main results |
16 |
(a) Give unadjusted estimates and, if applicable, confounder-adjusted estimates and their precision (eg, 95% confidence interval). Make clear which confounders were adjusted for and why they were included |
|
(b) Report category boundaries when continuous variables were categorized |
||
|
(c) If relevant, consider translating estimates of relative risk into absolute risk for a meaningful time period |
||
|
Other analyses |
17 |
Report other analyses done—eg analyses of subgroups and interactions, and sensitivity analyses |
|
Discussion |
||
|
Key results |
18 |
Summarise key results with reference to study objectives |
|
Limitations |
19 |
Discuss limitations of the study, taking into account sources of potential bias or imprecision. Discuss both direction and magnitude of any potential bias |
|
Interpretation |
20 |
Give a cautious overall interpretation of results considering objectives, limitations, multiplicity of analyses, results from similar studies, and other relevant evidence |
|
Generalisability |
21 |
Discuss the generalisability (external validity) of the study results |
|
Other information |
||
|
Funding |
22 |
Give the source of funding and the role of the funders for the present study and, if applicable, for the original study on which the present article is based |

Reviewer 5 Report
Comments and Suggestions for Authors
This is a very interesting and important study by Xue et al, to describe the importance of Neuronal Derived Exosome (NDE) as a potential biomarker to detect the disease severity of Parkinson’s disease. The article is well written, and experiments are systematically designed and conducted. The major strength of the article is that it draws an association between α-synuclein and Aβ42/Aβ40 ratio in the NDE from blood plasma of the PD patients
and the disease severity, but there are concerns on the qualitative and quantitative characterization of the NDEs enrichment from patient’s blood.
Specific comments:
1. It is widely known that the blood brain barrier permeability is increased in both Parkinson’s (https://doi.org/10.1038/s41467-023-39038-8 , https://doi.org/10.3389/fphys.2020.593026) and Alzheimer’s (https://doi.org/10.1002/alz.063948, https://doi.org/10.1016/j.nbd.2016.07.007) disease. To this end it is possible that due to leaky blood brain barrier, the PD patients may have more NDEs in their blood plasma compared to the Healthy Controls. Thus, in this context, it will be important to assess the
di􀆯erence in the abundance of NDEs in the plasma of HCs, PD-NC, PD-MCI and PDD groups.
2. Please provide detailed information such as catalog number and other relevant information on the Kaixianghongkang NDE extraction kit. As I am unable to find out any detail about this NDE extraction kit online. It will be helpful for readers as well as researchers in the same field.
3. As described in lines 129-132, the characterization data of neuronal-derived exosome (NDE) enrichment such as western blots, TEM micrographs should be provided to ensure the quality of enriched NDEs in the main or supplementary section of the article.
Author Response
Dear Editor, Dear reviewers
Thank you for your letter dated June 24. We were pleased to know that our work was rated as potentially acceptable for publication in Journal, subject to adequate revision. We thank the reviewers for the time and effort that they have put into reviewing the previous version of the manuscript. Their suggestions have enabled us to improve our work.
We uploaded the revised manuscript and our responses are given directly afterward in a different color (red).
We hope that the revised manuscript is accepted for publication in the Journal of Brain Sciences.
Sincerely, Xiaofan Xue.
Comment 1: It is widely known that the blood brain barrier permeability is increased in both Parkinson’s (https://doi.org/10.1038/s41467-023-39038-8 , https://doi.org/10.3389/fphys.2020.593026) and Alzheimer’s (https://doi.org/10.1002/alz.063948, https://doi.org/10.1016/j.nbd.2016.07.007) disease. To this end it is possible that due to leaky blood brain barrier, the PD patients may have more NDEs in their blood plasma compared to the Healthy Controls. Thus, in this context, it will be important to assess the
difference in the abundance of NDEs in the plasma of HCs, PD-NC, PD-MCI and PDD groups.
Response1: Thank you for pointing it out. Previous studies have shown that BBB disruption in PD, which is consistent with our result: NDE (which I mentioned as Neuronal extracellular vesicle (EV) below) α-syn was significantly higher in PD as compared to HC group (133.806±93.889 vs 69.328±45.475, p<0.001). Another paper said that BBB permeability increases in individuals with AD, which independent of amyloid pathology. This conclusion indicated that the level of amyloid protein is not parallel with the BBB permeability. As we all know, classical plasma biomarkers (decreased Aβ42 and Aβ42/Aβ40 ratio, increased p-tau181 and T-tau) have been incorporated into the diagnostic criteria for AD. The reason why we chose the Neuronal EV biomarkers is because it derived from the neuron in brain, which is thought to be close to the level of CSF. We hope there are some Neuronal EV biomarkers which are more sensitive than plasma ones. Unforunatly, the results of Neuronal EV levels of Aβ42, Aβ42/Aβ40 ratio, T-tau, p-tau181 and GFAP were not significantly different between PD and HC groups. However, the Neuronal EV T-tau level was significantly higher in the amnestic than the non-amnestic group. This is echo with the diagnostic criteria for AD which indicated that AD pathology involved in PD. future studies would be important for verifying our findings.
Comment 2: Please provide detailed information such as catalog number and other relevant information on the Kaixianghongkang Neuronal EV extraction kit. As I am unable to find out any detail about this NDE extraction kit online. It will be helpful for readers as well as researchers in the same field.
Response2: Thank you for this suggestion. We have dded relevant information with “specifically the Kaixianghongkang Neuronal EV extraction kit, C-005, Kaixianghongkang, Beijing, China” and “B-001, Kaixianghongkang, Beijing, China”.(Mentioned exactly where in the revised manuscript this change can be found-page3, line140, line144)
Comment 3: As described in lines 129-132, the characterization data of neuronal-derived exosome (NDE) enrichment such as western blots, TEM micrographs should be provided to ensure the quality of enriched NDEs in the main or supplementary section of the article.
Response 3: Thank you for pointing this out. Actually, all the biomarkers (plasma and Neuronal EVs) were measured using Single-molecule array and Chemiluminescence immunoassay respectively, not using the Western blotting. WB analysis was performed to verify the presence of typical exosome markers in Neuronal EVs. So we revised the sentence in abtract and also in the 2.4 part of Materials and Methods. By the way, we offer another supplementary figure about the details about the presence of typical EV markers in Neuronal EVs by WB. (Mentioned exactly where in the revised manuscript this change can be found-page1, line31; page4, line151-163; page9, line355-359)

Round 2
Reviewer 3 Report
Comments and Suggestions for Authors
Thank you for the rebuttal addressing my comments.
Reviewer 4 Report
Comments and Suggestions for Authors
Thank you for your reply, the article now has improved significantly.
Reviewer 5 Report
Comments and Suggestions for Authors
Dear Authors,
I have read the revised manuscript and find that all my concerns have been adequately addressed. The revised manuscript is now improved and I recommend that the article be accepted for publication.